SPMLMI: predicting lncRNA–miRNA interactions in humans using a structural perturbation method

Xu Mingmin 1
Chen Yuanyuan 2
Lu Wei 3
Kong Lingpeng 1
Fang Jingya 1
Li Zutan 1
Zhang Liangyun zlyun@njau.edu.cn 2
Pian Cong piancong@njau.edu.cn 2
1 College of Agriculture, Nanjing Agricultural University , Nanjing , Jiangsu , China
2 College of Sciences, Nanjing Agricultural University , Nanjing , Jiangsu , China
3 College of Life Sciences, Nanjing Agricultural University , Nanjing , Jiangsu , China
Qin Zhaohui
Electronic publication date: 2021 May 19
Publication date: 2021
Volume: 9
Electronic Location ID: e11426
Received 2021 Jan 26; Accepted 2021 Apr 18
Copyright: ©2021 Xu et al.
Copyright year: 2021
Copyright holder: Xu et al.
License: This is an open access article distributed under the terms of the Creative Commons Attribution License, which permits unrestricted use, distribution, reproduction and adaptation in any medium and for any purpose provided that it is properly attributed. For attribution, the original author(s), title, publication source (PeerJ) and either DOI or URL of the article must be cited.
License URL: https://creativecommons.org/licenses/by/4.0/

Keywords: lncRNA–miRNA interaction, Bilayer network, Structural perturbation, Structural consistency, Prediction algorithm

Funding: National Natural Science Foundation of China 11571173 Nanjing Agricultural University 050/804009 This work was supported by the National Natural Science Foundation of China (No. 11571173) and the Startup Foundation for Advanced Talents at Nanjing Agricultural University (No. 050/804009). The funders had no role in study design, data collection and analysis, decision to publish, or preparation of the manuscript.

==============================
Long non-coding RNA (lncRNA)–microRNA (miRNA) interactions are quickly emerging as important mechanisms underlying the functions of non-coding RNAs. Accordingly, predicting lncRNA–miRNA interactions provides an important basis for understanding the mechanisms of action of ncRNAs. However, the accuracy of the established prediction methods is still limited. In this study, we used structural consistency to measure the predictability of interactive links based on a bilayer network by integrating information for known lncRNA–miRNA interactions, an lncRNA similarity network, and an miRNA similarity network. In particular, by using the structural perturbation method, we proposed a framework called SPMLMI to predict potential lncRNA–miRNA interactions based on the bilayer network. We found that the structural consistency of the bilayer network was higher than that of any single network, supporting the utility of bilayer network construction for the prediction of lncRNA–miRNA interactions. Applying SPMLMI to three real datasets, we obtained areas under the curves of 0.9512 ± 0.0034, 0.8767 ± 0.0033, and 0.8653 ± 0.0021 based on 5-fold cross-validation, suggesting good model performance. In addition, the generalizability of SPMLMI was better than that of the previously established methods. Case studies of two lncRNAs (i.e., SNHG14 and MALAT1) further demonstrated the feasibility and effectiveness of the method. Therefore, SPMLMI is a feasible approach to identify novel lncRNA–miRNA interactions underlying complex biological processes.

Introduction

For a long time, our framework for understanding the nature of genetic programming in complex organisms was biased by the assumption that protein-coding genes are the majority of the bearers of genetic information. Researchers have recently abandoned their notions about the regulation of complex cellular mechanisms, shifting their focus to previously unexplored RNAs. We now know that RNAs are functionally versatile molecules directly involved in the regulation of many cellular processes, such as epigenetic control, gene transcription, translation, RNA transformation, chromosome organization, and genome defense as well as cell proliferation and developmental programs (Guil & Esteller, 2015; Morris & Mattick, 2014). In most instances, the key regulatory factors are non-coding RNAs (ncRNAs) (Huntzinger & Izaurralde, 2011; Rinn & Chang, 2012; Sabin, Delas & Hannon, 2013), which account for the vast majority of mammalian transcripts and range in length from 22 nucleotides to hundreds of kilobases. Although there are established principles that define classes of ncRNAs (e.g., tRNAs and miRNAs), it could be suggested that each ncRNA has a unique function (Cech & Steitz, 2014).

The first miRNA (microRNA) was reported in 1993 as a short RNA in Caenorhabditis elegans able to control the timing of developmental transitions by base pairing to partially complementary sequences in the 3′-UTR of its target mRNA (Lee, Feinbaum & Ambros, 1993; Wightman, Ha & Ruvkun, 1993). Today, miRNAs are well-known gene expression repressors (Béthune, Artus-Revel & Filipowicz, 2012; Braun, Huntzinger & Izaurralde, 2013; Huntzinger & Izaurralde, 2011), and are widely found in human and plant genomes (Krol, Loedige & Filipowicz, 2010; Voinnet, 2009). LncRNAs (long non-coding RNAs) are the most abundant ncRNAs; tens of thousands of human lncRNAs have been identified. However, the roles of most lncRNAs in cellular processes have yet to be determined, and we anticipate that many new functions will be discovered (Quinn & Chang, 2016; Chen et al., 2019a; Chen et al., 2017). Previous studies have concluded that lncRNAs could regulate gene expression patterns by biomolecular interactions, including lncRNA–protein, lncRNA–mRNA, and lncRNA–ncRNA interactions (Li et al., 2014). Yoon, Abdelmohsen & Gorospe (2014) reviewed functional interactions among miRNAs and lncRNAs with important roles in gene expression programs. We believe that detailed analyses of the interactivity between lncRNAs and miRNAs will enrich our understanding of the functions of cellular networks.

RNA–RNA interactions are rapidly emerging as one of the most important functional mechanisms involving ncRNAs (Lai & Meyer, 2016). Typically, the discovery of molecular interactions in biological network requires extensive experimental work (Clauset, Moore & Newman, 2008). Biological experiments are highly reliable for studies of biomolecular interactions; however, they are difficult to carry out on a large scale owing to the high costs and long execution times. An alternative to studying interactions between each pair of molecules in a huge and complex biological network is a predictive approach, in which prior interaction information is used to identify molecules that are most likely to be related, providing a basis for further experimental studies. In particular, computational prediction methods are useful for prioritizing candidate lncRNA–miRNA interactions on a large scale.

Most existing RNA–RNA interaction prediction methods are based on sequence information, including sequence conservation, seed region matching, site accessibility, and minimum free energy, and predict matching relationships between miRNAs, small RNAs, and small nucleolar RNAs with their targets (mRNAs and rRNAs) (Lai & Meyer, 2016; Quinn & Chang, 2016). Different from mRNAs and rRNAs, lncRNAs have substantial sequence variation and relatively low conservation, and they are involved in relatively complex molecular interactions. Accordingly, we cannot predict interactions between lncRNAs and miRNAs by sequence information alone.

As a primary type of competitive endogenous RNA (ceRNA), lncRNAs can act as miRNA sponges to weaken the effect of miRNAs on mRNAs; therefore, miRNAs play important roles in the molecular mechanism of lncRNAs (Salmena et al., 2011; Xu et al., 2015; Thomson & Dinger, 2016; Lv et al., 2020). ceRNA crosstalk appears to be closely related to miRNA expression levels, and the specificity of interactions may depend on the expression profile of miRNAs (Ala et al., 2013; Xu et al., 2015; Xu et al., 2016). In addition, some lncRNAs may be co-regulated in expression networks, suggesting that multiple lncRNAs may interact with specific miRNA clusters to regulate biological processes in a coordinated way (Li et al., 2015; Thomson & Dinger, 2016). It is therefore reasonable to expected the expression patterns of lncRNAs and miRNAs to be important information for predicting interactions.

How can known information be used to effectively predict lncRNA–miRNA interactions? This problem is essentially a linkage prediction problem for the interactions network (Lü & Zhou, 2011). Previous studies have focused on improving the accuracy of prediction methods, without investigating the extent to which lncRNA–miRNA interactions can be predicted. In this study, we refer to the model proposed by Lü et al. (2015) and use a structural consistency index to measure the predictability of interactions. Given the adjacency matrix of a network, its first-order matrix perturbation can describe the inherent properties of the network structure and can reflect the link predictability without prior knowledge of the network organization. That is, the consistency of structural features before and after the removal of a random component of the links reflects the predictability of a network. A model based on the structural perturbation method was developed to predict lncRNA–miRNA interactions. We refer to this model as SPMLMI (i.e., Structural Perturbation Method for predicting LncRNA–MiRNA Interactions).

This study makes several key contributions. First, we constructed an lncRNA–miRNA bilayer network composed of three kinds of information (lncRNA similarity, miRNA similarity, and the validated lncRNA–miRNA interactions). Second, we used the structural consistency index to measure the link predictability of multiple related networks and found that the structural consistency of the bilayer network was greater than that of any of single network, supporting the use of the bilayer network for link prediction. Furthermore, we employed SPMLMI to predict potential lncRNA–miRNA interactions. Last, experimental results showed that the newly developed method achieves good prediction performance using different datasets and that its generalizability is better than those of many previous methods. Overall, SPMLMI is an effective method for predicting unknown lncRNA–miRNA interactions.

Materials & Methods

Data collection

Our goal was to develop a method that is able to predict candidate interactions between lncRNAs and miRNAs. Three kinds of data were collected from various databases and a bilayer network of lncRNAs and miRNAs was constructed to represent the complex relationships. First, known lncRNA–miRNA interaction data were downloaded from the lncRNASNP database version 2.0 (http://bioinfo.life.hust.edu.cn/lncRNASNP) (Miao et al., 2018) and used to construct an lncRNA–miRNA network. Second, expression data for lncRNAs were downloaded from the NONCODE database (http://www.noncode.org/) (Fang et al., 2018) and used to calculate the expression similarity between lncRNAs and to construct the lncRNA similarity network. Additionally, miRNA expression data were downloaded from the miRmine database (https://guanfiles.dcmb.med.umich.edu/mirmine/index.html) (Panwar, Omenn & Guan, 2017) and used to construct the miRNA similarity network. Table 1 shows the data obtained from publicly available databases.

Table 1 Summary of data used in this study.

Database	Data types	Number of links/nodes	
lncRNASNP	lncRNA–miRNA interactions
(validated)	39366
(interactions/edges/links)	
NONCODE	lncRNA expression profiles	3,150
(lncRNA nodes)	
miRmine	miRNA expression profiles	262
(miRNA nodes)	

Construction of an lncRNA–miRNA bilayer network

An lncRNA–miRNA bilayer network containing two kinds of nodes (miRNAs and lncRNAs) and three types of relationships (miRNA–miRNA similarity relationship, lncRNA–lncRNA similarity relationship, as well as the known lncRNA–miRNA interaction relationship) was constructed.

To construct the lncRNA–miRNA interaction network, data were downloaded from the version 2.0 of the lncRNASNP database (Miao et al., 2018). This database contains known lncRNA–miRNA interactions with experimental validation collected from version 2.0 of the starBase (Li et al., 2014). Then, we define matrix LMnet as the known lncRNA–miRNA interaction network, where if lncRNA i and miRNA j are connected, the element LMnet = 1; otherwise, LMnet = 0. A total of 39,366 associations between 3,150 lncRNAs and 262 miRNAs were retained after removing duplicates. Figure 1B presents a simple example of lncRNA–miRNA interaction network construction.

Figure 1 Flowchart of SPMLMI.

(A) The lncRNA expression profiles were downloaded from the NONCODE database and used to calculate the lncRNA similarity network (LSnet). (B) Known lncRNA–miRNA interactions were downloaded from the lncRNASNP database and used to construct the lncRNA–miRNA interaction network (LMnet). (C) The miRNA expression profiles were downloaded from the miRmine database and used to construct the miRNA similarity network (MSnet). (C) Three networks were integrated to construct the lncRNA–miRNA bilayer network. (E) The perturbed network was obtained by using SPMLMI.

The lncRNA similarity network and miRNA similarity network were constructed based on expression profiles. First, expression profile data for miRNAs were downloaded from the miRmine database (Panwar, Omenn & Guan, 2017). The database contains data for 2588 miRNAs in 135 different human tissues and cell lines. After removing missing items and matching the miRNA IDs in the lncRNA–miRNA interaction network (LMnet),expression data for 262 miRNAs in 124 different human tissues and cell lines were retained. Second, lncRNA expression profiles were downloaded from the NONCODE database (Fang et al., 2018). Expression profiles of 171,814 transcripts in 16 different human tissues and eight cell lines were obtained. After the removal of duplicates and matching lncRNA IDs in LMnet, 3,150 lncRNA expression records for 24 different human tissues and cell lines were retained.

To describe the expression characteristics of ncRNAs in different tissues or cell lines in an unbiased fashion, the expression profiles were standardized before calculating pairwise similarity values. The Pearson correlation coefficient was used to measure the similarity of ncRNAs and the expression similarity matrix of lncRNAs and miRNAs was obtained. The similarity matrix represented the similarity network of these two types of ncRNAs, referred to as LSnet and MSnet, respectively, as shown in Figs. 1A and 1C. For a pair of RNAs, a higher correlation indicates greater similarity in expression, in general.

Finally, the above three networks are integrated to build a bilayer network. Specifically, the lncRNA–miRNA bilayer network can be represented by a block adjacency matrix A ∈ RN×N, (1) A=LSentLMnetLMnetTMSnet,

where N = 3,412 (the total number of nodes in the network, 3,150 lncRNAs and 262 miRNAs) and LMnetT is the transpose of LMnet.

Overall, the lncRNA–miRNA bilayer network was composed of the miRNA similarity network, lncRNA similarity network, and edges connecting the two networks. Figure 1 demonstrates the workflow for constructing the bilayer network and its matrix representation.

Structural consistency and structural perturbation method

Structural consistency was first proposed by Lü et al. (2015) and can be used to evaluate the likelihood of links prediction in complex network. It is defined as the consistency of network structural features before and after the elimination of partial associations at random. For the rationale and the specific derivation, refer to the supplementary material (File S1). In this study, this method was applied to the lncRNA–miRNA bilayer network A and its sub-networks (LSnet and MSnet) and the structural consistency of each network was evaluated separately.

Generally, the link prediction problem refers to the problem of estimating the probability of the existence of unobserved links according to known topological information. The network structure perturbation method involved in the structure consistency calculation process can be used to predict missing links (Lü et al., 2015). After obtaining the lncRNA-miRNA bilayer network (adjacency matrix, A) in ‘Construction of an lncRNA-miRNA bilayer network’, we derive the perturbation matrix A′ by the structural perturbation method. The prediction matrix A′ ˆ is then obtained by averaging t independent perturbation calculations. See the supplementary materials for more details (File S1). In this way, elements in the prediction matrix A′ ˆ can be interpreted as scores for each pair of nodes in the lncRNA–miRNA bilayer network. The scores in A′ ˆ determine the extent of all unobserved lncRNA–miRNA interactions, and we assume that a higher score corresponds to a greater likelihood of a potential interaction.

Evaluation metrics

To evaluate the performance of SPMLMI, k-fold cross-validation was performed based on verified lncRNA–miRNA interactions downloaded from the lncRNASNP database (Miao et al., 2018). In the cross-validation process, the original set was randomly divided into k equally sized subsets. Among these k subsets, one was used as the validation data for testing the model, and the others were used as the training data. Specifically, we randomly reset 1∕k known interaction entries of the training dataset to unknown, and prioritize the candidate interaction entries in the prediction matrix by SPMLMI scores. The known interactions are considered as positive, and the rest as negative. True positive rate is the proportion of overlap between the top 1∕k ranked candidate interactions with the known interactions. Then, the cross-validation was repeated k times, taking each of the k subsets as the validation data in turn. The cross-validation results are averaged to produce a single estimation.

The AUC (area under the receiver operating characteristic curve) was estimated to evaluate the performance of lncRNA–miRNA interaction prediction results. In particular, the TPR (true positive rate) and FPR (false positive rate) were calculated by varying the threshold and the ROC (receiver operating characteristic) curve and AUC value were obtained, where TPR and FPR represent the percentage of test samples above or below the given thresholds, respectively.

In addition, we applied SPMLMI to datasets analyzed by two additional methods, EPLMI (Huang, Chan & You, 2018) and SNFHGILMI (Fan, Cui & Zhu, 2020), for an objective comparison of performance.

Results

Structural consistency of different networks

We calculated the structural consistency of three datasets (i.e., the EPLMI dataset provided by Huang, Chan & You (2018), SNFHGILMI dataset provided by Fan, Cui & Zhu (2020), and SPMLMI dataset obtained in this study). Each group of datasets was composed of an miRNA similarity network (MSnet), an lncRNA similarity network (LSnet), and a bilayer network (Bilayer − net). We randomly selected 10% of the total links from each network as the perturbation set (note that the bilayer network contains two kinds of links). As shown in Fig. 2A, in three different datasets, the structural consistency values for the lncRNA–miRNA bilayer network (Bilayer − net, yellow bar) were 0.2413 ± 0.0015, 0.3036 ± 0.0011, and 0.2168 ± 0.0005, respectively. The structural consistency of these three bilayer networks (Bilayer − net) were all greater than that of the similarity network (where the blue bar is MSnet and the orange bar is LSnet in Fig. 2), indicating that considering more information can improve the structural consistency of the network. These results show that the construction of the bilayer network improves the inherent link predictability of the network and will be helpful for the prediction of “missing links” (i.e., newly established or novel interactions).

Figure 2 Experimental results on structural consistency, parameter t optimization and performance evaluation.

(A) Structural consistency of related networks. (B) Effect of the parameter t on model performance. (C) Performance of SPMLMI under different k-fold cross-validation frameworks.

Performance evaluation and parameter optimization

First, to evaluate the performance of our proposed SPMLMI for the prediction of lncRNA–miRNA interactions, we adopted 2-fold, 5-fold and 10-fold cross-validation frameworks. To avoid the bias of sample division in cross-validation, we repeated each experiment 10 times and took the average value. Figure 2C shows the corresponding ROC curve for each k-fold cross-validation. The average areas under the curves were 0.8617 ± 0.0013, 0.9512  ± 0.0034, and 0.9517 ± 0.0016 for 2-fold, 5-fold, and 10-fold cross-validation, respectively. These results indicate that k-fold cross-validation increase improves the performance of SPMLMI. There was no significant difference (paired t-test, p-value = 0.1997) between 5-fold and 10-fold cross-validation. Therefore, in subsequent experiments, we use 5-fold cross-validation to save time.

Second, to optimize t (i.e., the number of perturbations), we tuned the parameter from 1 to 16. Figure 2B shows boxplots of the variation of AUC values with increasing t. The AUC values increased as t increased; however, for values of t greater than or equal to 8, AUC was relatively stable. For simplicity, we set the parameter t to 8 for subsequent experiments.

Comparison with previous methods

Furthermore, we compared SPMLMI with the previously established EPLMI (Huang, Chan & You, 2018) based on the two-way diffusion algorithm and SNFHGILMI (Fan, Cui & Zhu, 2020) based on the heterogeneous graph inference method. Initially, we evaluated the same dataset (i.e., the SPMLMI dataset obtained in this study, see ‘Materials & Methods’ for details) by these three methods. Figure 3A shows the ROC curves for the prediction performance of these three models. The average AUC for SPMLMI was 0.9512 ± 0.0034, which was higher than the AUC values of 0.7999 ± 0.0014 for EPLMI and 0.5163 ± 0.0013 for SNFHGILMI, respectively. We also compared the methods using the original datasets. Using the EPLMI dataset provided by Huang, Chan & You (2018), the average AUC values for the three methods (SPMLMI, EPLMI, and SNFHGILMI) were 0.8767 ± 0.0033, 0.8417 ± 0.0017, and 0.5184 ± 0.0023, respectively (Fig. 3B). For the SNFHGILMI dataset provided by Fan, Cui & Zhu (2020), the average AUC values were 0.8653 ± 0.0021, 0.8570 ± 0.0012, and 0.9426 ± 0.0035, respectively (Fig. 3C).

Figure 3 Comparison with previous methods.

(A) Experimental results for the three methods on the SPMLMI dataset. (B) Experimental results for the three methods on the EPLMI dataset. (C) Experimental results for three methods on the SNFHGILMI dataset.

These results prove that the newly established SPMLMI has good performance and generalizability across datasets.

Case studies

Two lncRNAs, SNHG14 (UBE3A-ATS/NONHSAT041137) and MALAT1 (NONHSAT022132), with important biological functions were selected to validate the prediction performance of SPMLMI. SNHG14, also known as UBE3A-ATS, overlaps with the entire UBE3A gene and its paternal expression is thought to negatively regulate paternal UBE3A expression in neurons, thereby determining tissue-specific imprinting of UBE3A in the brain (Galiveti et al., 2014; Stanurova et al., 2016). MALA1 is located less than 70 kb from NCRNA00084 on chromosome 11q13.1 (Hutchinson et al., 2007) and is dysregulated in many diseases, such as alcohol-related diseases, endometrioid endometrial carcinoma, and non-small cell lung cancer (Ji et al., 2003; Kryger et al., 2012; Li et al., 2016). For each of this two lncRNAs in the interaction network, known interaction entries were randomly reset to unknown. We implemented SPMLMI using the reset data set and prioritized all candidate miRNAs according to their scores in the prediction matrix.

As shown in Table 2, among the top 10% (26) predicted SNHG14-related miRNAs, 18 lncRNA–miRNA interactions were confirmed by experimental validation (containing 17 of the 21 known interactions we collected from the lncRNASNP2 database Miao et al., 2018). It should be noted that the miRNA hsa-miR-126-3p (marked with an asterisk in Table 2), ranked 16th, was not collected from the lncRNASNP2 database (Miao et al., 2018) but was reported by an experimental study of transcriptome-wide Ago2:RNA interactions in human brain tissues using HITS-CLIP (i.e., CLIP-seq) (Boudreau et al., 2014). This information was retrieved from DIANA-LncBase v3 (Karagkouni et al., 2020). Several studies have shown that miR-126 directly targets the 3′-UTR of insulin receptor substrate-1 (IRS-1) (Ryu et al., 2011; Zhang et al., 2008; Zhou et al., 2013). Over-expression of miR-126 leads to downregulation of IRS-1 expression, and a consequent impairment in insulin signaling (Ryu et al., 2011). Moreover, the research results of Zhou et al. (2013) showed over-expression of miR-126 down-regulated IRS-1, suppressed AKT and ERK1/2 activation, CRC cells proliferation, migration, invasion, and caused cell cycle arrest. Therefore, lncRNA SNHG14, as an interacting molecule of miR-126, may play a potential role in regulating cell metabolism and cell cycle.

Table 2 Prediction results for the top 10% SNHG14-related miRNAs ranked by prediction scores.

Rank	MiRNAs	Reference
(PMID)	Rank	MiRNAs	Reference
(PMID)	
1	hsa-miR-425-5p	23313552	14	hsa-miR-32-5p	24668909;
28030800;
22291592	
2	hsa-miR-181b-5p	23313552;
28030800;
22012620	15	hsa-miR-92a-3p	24668909;
28030800;
22291592	
3	hsa-miR-340-5p	24668909;
23313552	16	hsa-miR-126-3p ∗	24389009	
4	hsa-miR-25-3p	24668909;
28030800	17	hsa-miR-181c-5p	23313552;
28030800;
22012620	
5	hsa-miR-363-3p	24668909;
28030800;
22291592	18	hsa-miR-382-5p	23313552	
6	hsa-miR-543	23313552;
28030800	19	hsa-miR-551a	Unconfirmed	
7	hsa-miR-433-3p	22012620	20	hsa-miR-200c-3p	Unconfirmed	
8	hsa-miR-136-5p	22012620	21	hsa-miR-187-3p	Unconfirmed	
9	hsa-miR-181a-5p	23313552;
28030800;
22012620	22	hsa-miR-378b	Unconfirmed	
10	hsa-miR-181d-5p	23313552;
28030800;
22012620	23	hsa-miR-100-5p	Unconfirmed	
11	hsa-miR-376c-3p	24668909;
22100165	24	hsa-miR-758-3p	Unconfirmed	
12	hsa-miR-206	24906430;
22012620	25	hsa-miR-10b-5p	Unconfirmed	
13	hsa-miR-92b-3p	24668909;
28030800;
22291592	26	hsa-miR-300	Unconfirmed	

Similarly, for MALAT1, we examined the candidate miRNAs ranked by prediction scores. The results showed that the top 10% of predicted MALAT1-related miRNAs were all verified in published studies (Table S1). The reason is most likely that lncRNA MALAT1 has more known interaction information (containing 113 known interactions collected from the lncRNASNP2 database (Miao et al., 2018)). Therefore, we examined more predicted candidate miRNAs and found that a total of 106 were verifiable. Additionally, in a genome-wide analysis of miRNA–mRNA interactions in marrow stromal cells, the interaction between lncRNA MALAT1 and miRNA hsa-miR-487b-3p was confirmed (Balakrishnan et al., 2014); this candidate miRNA was ranked 91st (Table S1) based on prediction scores. The experimental study by Xi et al. (2013) demonstrated that miR-487b directly targets gene Wnt5a. The protein Wnt5a is an important activator of the Wnt/Ca2+ signaling pathway, which plays an important role in embryonic development, cell differentiation, inflammation and other physiological processes (De, 2011). Because of sharing miRNA miR-487b with Wnt5a gene, lncRNA MALAT1 may also be involved in this complex signaling network.

The above results suggest that SPMLMI has good prediction ability and can be used to screen candidate target miRNAs for a given lncRNA.

Discussion

The identification of associations among biomolecules has important implications for our understanding of biological processes. In view of the complexity of intermolecular interaction networks, it is necessary to establish a prediction model that makes full use of prior information. Lü et al. (2015) studied the link predictability of complex networks and proposed the structural perturbation method for predicting “missing links” in networks. Zeng et al. (2018) applied this algorithm to biological networks. They constructed a bilayer network and predicted unknown miRNA–disease associations by the perturbation of specific diseases in the network, achieving good results. We further extended the structural perturbation method; the key difference in our study was the application of structural perturbation to the entire known lncRNA–miRNA interaction network, without focusing on a specific ncRNA (lncRNA or miRNA).

The ceRNA hypothesis (Salmena et al., 2011) predicts that similarity in the expression profile of a pair of lncRNAs is beneficial if they interact with the same miRNA. In our study, the successful prediction of lncRNA–miRNA interactions based on ncRNA expression profile similarity seems to support this hypothesis; however, the reality is more complex for a few reasons. First, the ncRNA expression profile data used in this study were collected from different human tissues and cell lines (see ‘Construction of an lncRNAmiRNA bilayer network’). The similarity network reflects the generalized expression pattern of a pair of ncRNAs, rather than their expression pattern in a specific cellular environment. Second, an interaction between an lncRNA and miRNA may involve other molecules. Because lncRNAs are abundant in vivo and have various interactions with other molecules (Liu et al., 2017; Liu et al., 2015). Therefore, the indirect interactions of lncRNAs and miRNAs via other molecules, such as protein complexes, cannot be ignored.

Based on our results, we suggest that the network structure perturbation method has two main advantages over other approaches: (1) it can make full use of prior information in the network (2) and it captures network topology features well. However, the method has limitations. First, it requires prior knowledge of network nodes and links to achieve good performance. Second, our predicted candidate interaction pairs may contain indirect relationships involving other molecules (e.g., proteins, nucleic acids, etc.). The above summary of advantages and disadvantages are aspects of the method that should be noted, and also point out a direction we need to further research in the future. In addition, previous researchers have many excellent research work on miRNA-disease associations (Chen et al., 2019b) as well as lnRNA-disease associations (Chen & Yan, 2013). The studies of ncRNA-disease association by Chen & Yan (2013) and Chen et al. (2019b) inspired us to expand ncRNA-disease association studies by combining a priori information and reliable lncRNA-miRNA prediction information in our future research work.

Overall, the approach is valuable for large biomolecular networks and the generalized prediction of unknown links in the network or for analyses of specific molecules in a network. The method can be adapted according to the specific research aims.

Conclusions

We propose a framework named SPMLMI based on a bilayer network to predict interactions between lncRNAs and miRNAs. The bilayer network integrates similarity information for two types of ncRNA molecules and known interaction information. By using SPMLMI, we obtained a perturbation matrix and corresponded each matrix element to the correlation score between pairs of nodes in the network. In this way, missing links (i.e., unknown lncRNA–miRNA interactions) in the network are assigned correlation scores, where high scores indicate probable interactions. SPMLMI was feasible and effective for predicting lncRNA–miRNA interactions and represents a substantial improvement over SPMLMI.

Extensive regulatory functions of ncRNAs have been confirmed. Based on expression data, our method infers potential interactions between lncRNAs and miRNAs, providing a basis for further functional analyses. However, other regulatory interactions, such as lncRNA–mRNA interactions, miRNA–mRNA interactions, and ncRNA–protein interactions, have not been considered and will be a focus of future research.

Supplemental Information

Supplemental Information 1 Supplemental Information .

Click here for additional data file.

Supplemental Information 2 Prediction results for the top 120 MALAT1-related miRNAs ranked by prediction scores

Click here for additional data file.

Additional Information and Declarations

Competing Interests

Author Contributions

Data Availability

The authors declare there are no competing interests.

Mingmin Xu conceived and designed the experiments, performed the experiments, analyzed the data, prepared figures and/or tables, authored or reviewed drafts of the paper, and approved the final draft.

Yuanyuan Chen and Cong Pian conceived and designed the experiments, performed the experiments, analyzed the data, authored or reviewed drafts of the paper, and approved the final draft.

Wei Lu, Jingya Fang and Zutan Li analyzed the data, authored or reviewed drafts of the paper, and approved the final draft.

Lingpeng Kong performed the experiments, analyzed the data, prepared figures and/or tables, authored or reviewed drafts of the paper, and approved the final draft.

Liangyun Zhang conceived and designed the experiments, authored or reviewed drafts of the paper, and approved the final draft.

The following information was supplied regarding data availability:

The codes and datasets are available at GitHub: https://github.com/xumingmin/SPMLMI.git.

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
