# Peer review of "SPMLMI: predicting lncRNA–miRNA interactions in humans using a structural perturbation method"

_PeerJ, doi:10.7717/peerj.11426_

## Round 0.1 · original submission · Major Revisions

The reviewers' comments are mostly positive. Please revise your manuscript based on the reviewer's suggestions.

Reviewer 1 ·

Basic reporting

In this study, Xu et al. developed a computational method, called SPMLMI, to predict IncRNA–miRNA interactions in humans. This study is well designed and the manuscript is well organized.

Experimental design

Why do the authors take two lncRNAs SNHG14 and MALAT1 as case studies?

Validity of the findings

The authors should describe in more detail how the predicted interactions are judged as true values in cross-validation.

Additional comments

1. The authors should discuss the limitations of their method and describe the scope of application and considerations for using the method accordingly, so that the reader can use or evaluate the method.
2. The data collection section needs to be described in more detail.
3. Why do the authors take two lncRNAs SNHG14 and MALAT1 as case studies?
4. Could you give some discussions how to use predicted lncRNA-miRNA interactions to predict miRNA-disease associations and lncRNA-disease associations as the future direction of this work (PMIDs: 24002109 and 29045685)?
5. The authors should describe in more detail how the predicted interactions are judged as true values in cross-validation.
6. You should revise your English writing carefully and eliminate small errors in the paper to make the paper easier to understand.
7. The sentence of " However, the roles of most lncRNAs in cellular processes have yet to be determined, and we can safely predict that many new functions will be discovered. “should have references of paper with PMIDs of 27345524 and 30247501.

Reviewer 2 ·

Basic reporting

In this work, the authors develop a miRNA-lncRNA interaction predictive method based on network analysis, which considering the expression similarity between miRNAs and the similarity between lncRNAs respectively. Compared with previous ceRNA binary network, the added links between same node type improve the correlation of genes’ expression. Additionally, the higher AUC of SPMLMI comparing other 2 method indicates the higher accuracy of this approach. However, there are still several problems need to be explained.

Experimental design

no comment

Validity of the findings

no comment

Additional comments

Major Points:
1. The description of method is brief. Authors would better describe the interaction score and structural perturbation method in detail.
2. There is no consistent threshold in SPMLMI, such as significance or the top 10% with the highest score, for the prediction method. In the two application cases, the authors focused on the top 20 and 120 prediction results respectively.
3. It is mentioned in this paper that lncRNA has high tissue specificity, but it’s not reflected in the process of network construction.
4. Two case studies identified important lncRNA targeted miRNA, there was no further verification and biological explanation for their predicted interactions.

---

## Round 0.2 · Minor Revisions

Please address the comments of the second reviewer. Thank you.

Reviewer 1 ·

Basic reporting

It could be accepted for publication.

Experimental design

Good design

Validity of the findings

reliable findings

Additional comments

could be accepted

Reviewer 2 ·

Basic reporting

The authors have addressed the majority of my questions. The following papers (PMID: 32193291; PMID: 27365046; PMID: 26304537) should be cited which are closely associated with this work.

Experimental design

no other comments

Validity of the findings

no other comments

Additional comments

The authors have addressed the majority of my questions. The following papers (PMID: 32193291; PMID: 27365046; PMID: 26304537) should be cited which are closely associated with this work.

---

## Round 0.3 · accepted · Accept

Thanks for addressing the reviewers' comments.